# T-Cell-Dominated Immune Response Resolves Protracted SARS-CoV-2 Infection in the Absence of Neutralizing Antibodies in an Immunocompromised Individual

**DOI:** 10.3390/microorganisms11061562

**Published:** 2023-06-12

**Authors:** Till Bunse, Nina Koerber, Hannah Wintersteller, Jochen Schneider, Alexander Graf, Aleksandar Radonic, Andrea Thuermer, Max von Kleist, Helmut Blum, Christoph D. Spinner, Tanja Bauer, Percy A. Knolle, Ulrike Protzer, Eva C. Schulte

**Affiliations:** 1Institute of Virology, School of Medicine, Technical University of Munich, 81675 Munich, Germany; 2Institute of Virology, Helmholtz Munich, Trogerstrasse 30, 81675 Munich, Germany; 3Institute of Molecular Immunology and Experimental Oncology, School of Medicine, Technical University of Munich, 81675 Munich, Germany; 4Department of Internal Medicine II, University Hospital Rechts der Isar, School of Medicine, Technical University of Munich, 81675 Munich, Germany; 5Laboratory for Functional Genome Analysis, Gene Center, LMU Munich, 81377 Munich, Germany; 6Method Development, Research Infrastructure & IT (MFI), Robert-Koch Institute (RKI), 13353 Berlin, Germany; 7Department of Mathematics and Computer Science, Freie Universität (FU) Berlin, 14195 Berlin, Germany; 8Project Groups, Robert-Koch Institute (RKI), 13353 Berlin, Germany; 9German Center for Infection Research (DZIF), Munich Partner Site, 81675 Munich, Germany; 10Department of Psychiatry, University Hospital, LMU Munich, 80336 Munich, Germany; 11Institute of Psychiatric Phenomics and Genomics, University Hospital, LMU Munich, 80336 Munich, Germany

**Keywords:** immunocompromise, SARS-CoV-2, variants, protracted infection, B-and T-cell immunity

## Abstract

Immunocompromised individuals are at higher risk of developing protracted and severe COVID-19, and understanding individual disease courses and SARS-CoV-2 immune responses in these individuals is of the utmost importance. For more than two years, we followed an immunocompromised individual with a protracted SARS-CoV-2 infection that was eventually cleared in the absence of a humoral neutralizing SARS-CoV-2 antibody response. By conducting an in-depth examination of this individual’s immune response and comparing it to a large cohort of convalescents who spontaneously cleared a SARS-CoV-2 infection, we shed light on the interplay between B- and T-cell immunity and how they interact in clearing SARS-CoV-2 infection.

## 1. Introduction

Following its worldwide spread, intensive research efforts have led to unprecedented progress in our understanding of severe acute respiratory syndrome coronavirus 2 (SARS-CoV-2) and the diverse outcomes of the coronavirus infectious disease (COVID-19) it causes. There is an urgent need to understand virus and symptom persistence and adapt clinical management for diverse patient populations. As one step towards improving clinical management, extensive research has been conducted to understand how the host’s immune response controls infection and symptoms (e.g., [1,2,3]). 

Immunosuppressed individuals with COVID-19 constitute a largely under-represented group in these studies. When compared to immunocompetent people, immunocompromised people have a higher overall mortality rate as well as a higher rate of complications [4]. Additionally, in some cases, prolonged viral shedding further complicates clinical management [5,6]. However, both immunosuppression itself and prolonged viral shedding can be associated with very different courses of disease. While some immunocompromised individuals remain asymptomatic throughout the course of infection and eventually clear it [6], several instances have been reported where they were unable to clear the virus after prolonged periods of infection and eventually succumbed to COVID-19 [5,7]. The reasons for these differential presentations are currently only partially understood.

Here, we report a case of prolonged viral persistence in an immunosuppressed individual who eventually cleared the infection in the absence of a neutralizing SARS-CoV-2 antibody response. By conducting a more detailed examination of this individual’s immune response, following the disease course over almost two years, and comparing it to that of convalescents who spontaneously cleared an infection, we hope to shed light on the interplay of B- and T-cell immunity and how they interact to clear the infection.

## 2. Case Report

During the first wave of the pandemic, a 69-year-old woman was admitted to the emergency department of the University Hospital München Rechts der Isar, Munich, Germany, reporting a four-day history of fever, cough, and loss of taste. The initial suspected diagnosis of COVID-19 was confirmed by a positive SARS-CoV-2 PCR test and computer tomography revealing bilateral (viral) pneumonia. Thirteen months prior to this event, the patient was diagnosed with marginal zone lymphoma (stage IV, pancytopenia and autoimmune haemolytic anaemia). At the time of primary SARS-CoV-2 infection, she was in partial remission after being treated with bendamustine in combination with the monoclonal anti-CD20 antibody rituximab (last treatment seven months prior to infection) followed by obinutuzumab, another monoclonal anti-CD20 antibody (two cycles, last cycle five months prior to infection). At the time of admission, monoclonal gammopathy of unknown significance (MGUS, IgM lambda) and a secondary antibody deficiency were documented. 

The patient remained positive for SARS-CoV-2 RNA, and her symptoms persisted for a month, at which point a 10-day course of remdesivir (RDV) with intravenous immunoglobulins (IVIGs) was administered. Although the patient subsequently recovered fully in clinical terms, she remained SARS-CoV-2-PCR-positive and antibody-negative for SARS-CoV-2. Thus, another regime of 7 days of RDV in combination with convalescent plasma was administered starting at 90 days after infection. This resulted in a decrease in viral load to below the limit of detection, and the patient was discharged a little over three months after the initial infection, although antibodies against SARS-CoV-2 remained undetectable (Figure 1A).

During a routine screening 2 months later (day 162), the patient again tested positive for SARS-CoV-2 RNA, and her test positivity persisted for more than 6 months (Figure 1A). Neither anti-SARS-CoV-2-spike- nor nucleocapsid-specific IgG or IgM or virus-neutralizing antibodies (NAb) were detected at any time during this period (day 103 to day 327) (Figure 1A). Although mild symptoms such as anosmia, fever, and cough returned for a short period of time, the patient was mostly feeling well enough to quarantine at home on her own during this time. She last tested PCR-positive for SARS-CoV-2 on day 327 (Figure 1A).

On day 364, day 406, and day 533 post-infection with SARS-CoV-2, the patient received single vaccinations of the mRNA vaccine BNT162b2 (Comirnaty^®^, BioNTech-Pfizer), in line with the recommendations of the German national authorities (Figure 1A). Vaccine-induced IgG or NAb was not detectable at any point after the vaccinations (Figure 2A). Overall, the clinical course was relatively mild, but she was still reporting post-COVID symptoms, such as difficulty concentrating and gait instability, at the end of the observational period (day 619) without a strong trend towards improvement.

## 3. Results

In order to gain insights into this unusual but clinically highly relevant presentation of a COVID-19 case and to understand the determinants of long-term SARS-CoV-2 persistence and control without detectable antibody responses, we performed in-depth virus and host immune profiling.

### 3.1. SARS-CoV-2 Whole-Genome Sequencing

SARS-CoV-2 whole-genome sequences from nasopharyngeal swabs collected between day 13 and day 70 during the disease course showed that the patient was initially infected with the dominant viral strain at the time (Pangolin lineage B.1). Over the course of 58 days of continued SARS-CoV-2 PCR-positivity, several variants appeared to increase in frequency. These included both synonymous (2497A > T/G744G (ORF1ab); 11629T > C/Y3788Y (ORF1ab)) and non-synonymous variants (1121C > T/P286S (ORF1ab); 6629c > T/L2122F (ORF1ab); 13562A > T/K4433I (ORF1ab)) (Figure 1B). However, none of these were located in regions of the genome that were likely subject to immune-escape-related variants (i.e., the spike gene). This must be viewed in light of a lacking antibody response and, thus, a lack of antibody-mediated immune pressure. Similar to the situation observed in another immunocompromised host with persistent SARS-CoV-2 infection [7], variants of highly fluctuating frequency were also observed (e.g., 14898C > A/G4878G (ORF1ab); 28253C > T/F120F (ORF8)) (Figure 1B). Viral sequencing beyond day 70 did not yield high-quality sequences and, therefore, was not included in the analysis. 

### 3.2. Profiling of SARS-CoV-2-Specific Immunity

Based on the reasoning that in the absence of B-cell responses, virus-specific T-cell responses are decisive for virus control and clearance after infection, we performed a four-colour IFN-γ /IL-2/TNF/GzmB multiplex Fluorospot assay to determine the numbers of circulating cytokine-secreting cells reactive to SARS-CoV-2-spike-S1- and S2-specific peptide pools after vaccination with the BNT162b2 mRNA vaccine. 

Two weeks after the first vaccination (vaccination #1), we detected high numbers of spike-reactive IFN-γ-, IL-2-, and, in particular, TNF-secreting cells (Figure 2B). S1-reactive IFN-γ-secreting cells were further increased two weeks after the second vaccination (vaccination #2), while the numbers of S2-reactive IFN-γ- and S1/S2-reactive IL-2- and TNF-secreting cells were stable or even decreased. Three months after booster vaccination (vaccination #3), persistently high numbers of S1-reactive IFN-γ- and S1-/S2-reactive TNF-secreting cells were detected, while spike-reactive IL-2-secreting cells further declined (Figure 2B). Spike-reactive GzmB-secreting cells were absent at all time points. 

Interestingly, the patient showed substantially higher numbers of spike-reactive IFN-γ-, IL-2-, and TNF-secreting cells than individuals in the “healthy” control group (HC) of non-immunocompromised convalescents who spontaneously cleared SARS-CoV-2 infection and subsequently received three COVID-19 vaccinations of the BNT-162b2 mRNA vaccine (Figure 2B) (HC data partially published in [8]).

To further characterize the spike-reactive T-cells in the immunocompromised patient, we performed flow-cytometry-based surface and intracellular cytokine staining (ICS) of PBMCs three months after the third vaccination. We detected a particularly high frequency of total CD8+ T-cells (71.6%) but a low frequency of total CD4+ T-cells (23.0%) (Figure 2C, left panel). In comparison to HC, the absolute CD4 T-cell counts were decreased and the CD8 T-cell counts increased (Appendix A). At 3 months after the third vaccination, low frequencies of CD20+ B-cells became detectable again, marking 24 months after αCD20 therapy (Appendix A). Strikingly, ICS revealed high frequencies of S1-reactive, bi-functional IFN-γ-/TNF-producing effector CD8 but not CD4 T-cells (Figure 2C, middle and right panels). Overall, 18.3% of the S1-reactive CD8 T-cells stained positive for IFN-γ and TNF, but no S2-reactive CD8 T-cells were detected (Figure 2C and Appendix A). Surface staining elicited a high proportion of CD45RA+ CD8 T-cells expressing the fractalkine receptor CX3CR1 (57.0% of CD8 T-cells) compared to HC, indicating high frequencies of memory CD8 T-cells with cytotoxic effector function (Figure 2D).

## 4. Discussion

Understanding COVID-19 disease courses and immune responses in immunocompromised individuals is of the utmost importance. However, due to highly variable individual courses, levels of immune suppression, and time points of infection with SARS-CoV-2, many reports focus on individual cases. Here, we studied the immune response of an individual with a protracted SARS-CoV-2 infection after anti-CD20 therapy for over two years and found very high numbers of SARS-CoV-2-specific CD8 T-cells after vaccination but could not detect neutralizing antibodies. A comparison of B- and T-cell immunity to that of convalescents who spontaneously cleared the infection and received vaccinations allowed us to determine the relative strength of the patient’s immune response.

While some studies reported that the treatment of SARS-CoV-2-infected immunocompromised individuals with convalescent plasma but not RDV led to the emergence of potential immune escape variants in the spike gene [5,7], we and other researchers [6] did not observe the development of SARS-CoV-2 immune escape variants under these conditions. Over the period of more than two months of persistent SARS-CoV-2 infection, no novel spike gene variants emerged and became dominant, either spontaneously or after therapy with convalescent plasma or RDV, adding to the existing evidence showing that the use of convalescent plasma does not necessarily elicit sufficient immune pressure to select viral variants [6,7].

In our case, it seemed that host factors contributed more than viral factors to infection control and eventual viral clearance. Three months after the initial infection, the patient became SARS-CoV-2-negative for several weeks. Although multiple replication waves after a single SARS-CoV-2 infection were described in the immunocompromised host [6], we could not exclude reinfection in our patient but deemed this unlikely not least due to the low local incidence (around 35 cases/100,000 inhabitants/7 days) at the time of renewed SARS-CoV-2 positivity [9]. 

Despite the complete lack of development of SARS-CoV-2-specific IgG and IgM antibodies after infection and a full mRNA vaccination regime, the persistent SARS-CoV-2 infection was eventually controlled. This coincided with a dominant spike-S1-specific CD8 T-cell response, which was markedly higher than that in the group of vaccinated convalescents analysed at comparable time points after vaccination. This indicated a pre-existing memory T-cell response established during prolonged SARS-CoV-2 infection in this anti-CD20-treated patient. The high numbers of IFN-γ- and TNF-producing CD8 T-cells at three months after booster vaccination may have compensated for the absence of virus-specific antibodies and were likely the key determinant of virus control and eventual clearance, albeit that humoral immune responses were missing in this anti-CD20-treated individual. 

The differences in the detected frequencies of spike-S2-specific cells in the Fluorospot and the ICS assays might be explained by the different underlying assay principles. Both assays were performed with antigen re-stimulated PBMCs. In this case, upon stimulation, the activated cells start to produce cytokines. Using the Fluorospot approach, a discrimination between cytokine-secreting CD4 and CD8 T-cells is not possible unless a selection for the respective T-cells is performed in advance. Therefore, the detected IFN-γ- and/or TNF-secreting cells in the Fluorospot assay might stem not only from the T-cell population but also from other antigen-specific cytokine-secreting cells within PBMCs (i.e., NK cells).

Our observation is in line with other reports of patients undergoing anti-CD20 therapy for hematologic cancers or multiple sclerosis, in which the importance of a robust CD8 T-cell response, as a fundamental prerequisite for SARS-CoV-2 control and clearance, and relatively mild courses of the disease and expanded CD8 T-cell responses upon mRNA SARS-CoV-2 vaccination were highlighted [10,11,12,13,14,15]. In-depth flow cytometric analysis recapitulated reports of high numbers of CX3CR1-positive memory T-cells with effector function [16] in response to SARS-CoV-2 infection in immunocompromised individuals [11]. The strength of our study is that we were able to compare the B- and T-cell responses in our patient to those of a large cohort of convalescents who spontaneously cleared a SARS-CoV-2 infection and were vaccinated according to an identical vaccination scheme.

With regard to the clinical management of immunocompromised individuals with COVID-19, the case presented herein demonstrates the importance of virus-specific T-cell responses in controlling and clearing SARS-CoV-2 infection. In seronegative patients, even after complete vaccination regimes, a more detailed assessment of the T-cell response (e.g., by IFN-γ ELISpot) is warranted to understand the full range of immunity to SARS-CoV-2 in immunocompromised individuals.

## 5. Materials and Methods

### 5.1. SARS-CoV-2 Whole-Genome Sequencing

SARS-CoV-2 genomes were sequenced from RNA obtained from nasopharyngeal swabs. Sequencing libraries were prepared using the CleanPlex^®^ SARS-CoV-2 Next-Generation Sequencing Panel in line with manufacturer’s instructions and sequenced as paired-end (2 × 150 bp) runs with an ISeq100 (Illumina, San Diego, CA, USA). The sequenced reads were demultiplexed and aligned to the SARS-CoV-2 reference genome (NC 045512.2) with bwa-mem, and consensus sequences were assembled using iVar. “High-quality” sequences were defined as having ≤ 10% uncalled bases and ≥ 80% coverage across the genome. Only variants that reached a minor allele frequency (MAF) > 40% at some point during the observational period were included in the analysis. All sequence analyses were performed using the Integrated Genome Viewer (IGV) and R (version 4.0.2).

### 5.2. Isolation of Peripheral Blood Mononuclear Cells (PBMCs)

Blood was drawn with the Vacutainer CPT™ System and dispensed into sodium citrate CPT tubes (Becton Dickinson Biosciences, San Jose, CA, USA), and the tubes were mixed five times before being stored upright at room temperature. PBMCs were isolated within two hours of blood collection, as described previously [8]. For counting, the cells were resuspended in CTL Test Medium (CTL Europe GmbH, Bonn, Germany), and 10 µL of the cell suspension was diluted 1:2 with CTL Live/Dead Cell-Counting Dye (CTL-LDC Live/Dead Cell-Counting Kit, CTL Europe, Bonn, Germany). Then, 10 µL of the stained cell suspension was pipetted into the counting chamber, and cell counting was performed with an ImmunoSpot Ultimate UV Image analyser (CTL Europe, Germany).

### 5.3. Multi-Parameter IFN-γ /IL-2/TNF/Granzyme B Fluorospot Assay

Human IFN-γ/IL-2/TNF/Granzyme B four-colour Fluorospot assays (CTL Europe GmbH, Germany) were performed using freshly isolated peripheral blood mononuclear cells (PBMCs) according to the manufacturer’s instructions, as described previously [8]. The PBMCs were stimulated for 22 h with 1.0 µg/mL of overlapping peptide pools (15 mers overlapping by 11 aa) of the SARS-CoV-2 spike protein (PepMix™ SARS-CoV-2 (PM-WCPV-S), consisting of two peptide pools, i.e., S1 and S2, with 158 and 157 peptides, respectively (JPT Peptide Technologies, Berlin, Germany). Fluorospot plates were scanned and analysed using an automated reader system (ImmunoSpot Ultimate UV Image analyser/ImmunoSpot 7.0.17.0 Professional DC Software, CTL Europe GmbH, Germany). The counting of spot-forming cells (SFCs) was performed by adjusting the sensitivity, background balance, and the gates for the spot size using CTL software. All counts were reviewed and certified by a second person in a quality control process. The final results are represented as spot-forming cells (SFCs) per 1 × 10^6^ PBMCs. Positive reactivity to the experimental stimulatory agents was determined when the spot count for the antigen-stimulated cells was more than twice the spot count in the unstimulated (background) wells.

### 5.4. Flow-Cytometry-Based Ex Vivo Detection of SARS-CoV-2-Reactive T-Cells

Subsequent to co-stimulation with anti-CD28 (1.0 μg/mL; BD Biosciences, Cat# 340975) to ensure effective T-cell stimulation, 1 × 106 freshly isolated PBMCs were stimulated with 1.0 μg/mL of the aforementioned SARS-CoV-2 spike S1 and S2 peptide pools (JPT Peptide Technologies). After 1 h of incubation (37 °C, 5% CO_2_), 10 μg/mL Brefeldin A (Sigma-Aldrich, Burlington, MA, USA) was added to the cell suspension and incubated for 4 h at 37 °C in 5% CO_2_. 

For phenotypic characterization, the stimulated PBMCs were labelled with the LIVE/DEAD™ Fixable Blue Dead Cell Stain Kit (Thermo Fisher Scientific, Waltham, MA, USA) in a total volume of 100 μL for 30 min at 4 °C in the dark and washed twice with 200 μL FACS buffer (BD Biosciences). After centrifugation (560× *g*, 4 °C, 5 min), the PBMCs were stained with BV650 mouse anti-human CD20 (0.5 µg/mL, BioLegend, San Diego, CA, USA, Cat# 302335), EF450 mouse anti-human CD4 (0.5 µg/mL, eBioscience, Thermo Scientific, Cat# 48-0047-42), FITC mouse anti-human CD8 (0.5 µg/mL, Thermo Scientific, Cat# 11-0089-42), BV510 mouse anti-human CD45RA (0.25 µg/mL, BioLegend, Cat# 304142), APC Cy7 mouse anti-human CD45RO (2 µg/mL, Bio-Legend, Cat# 304228), PE mouse anti-human PD-1 (0.25 µg/mL, BioLegend, Cat# 329905), PerCP Cy5.5 mouse anti-human CX3CR1 (2 µg/mL, BioLegend, Cat# 341614), and BV605 mouse anti-human CD40L CD54 in a total volume of 50 µL including a brilliant violet buffer (BD Pharmingen Stain Buffer, BD Biosciences) for 30 min at 4  °C in the dark. 

In a separate second panel, surface marker and intracellular cytokine staining was performed as described previously [8]. Stimulated PBMCs were labelled with the LIVE/DEAD™ Fixable Blue Dead Cell Stain Kit (Thermo Fisher Scientific) in a total volume of 100 μL for 30 min at 4 °C in the dark and washed twice with 200 μL FACS buffer (BD Biosciences). After centrifugation (560× *g*, 4 °C, 5 min), the PBMCs were fixed for 20 min at 4 °C in the dark in 100 μL of an intracellular fixation buffer (Intracellular Fixation Buffer, Thermo Fisher Scientific, USA). After two wash steps with 200 μL/well Perm/Wash solution (Cytofix/Cytoperm Kit; BD Biosciences) and a centrifugation step (710× *g*, 4 °C, 5 min), the PBMCs were stained with BV510 mouse anti-human CD3 (1.0 μg/mL, BioLegend, Cat# 344828), EF450 mouse anti-human CD4 (0.5 μg/mL, eBioscience, Thermo Scientific, Cat# 48-0047-42), ECD mouse anti-human CD8 (0.2 μg/mL, Beckman Coulter, Brea, CA, USA, Cat# 737659), Al700 mouse anti-human IFN-γ (0.1 μg/mL, BD Bioscience, Cat# 557995), BV785 mouse antihuman TNF (5.0 μg/mL, BioLegend, Cat# 502948), and FITC rat anti-human IL-2 (3.1 μg/mL, eBioscience, Thermo Scientific, Cat# 11-7029-42) in a total volume of 80 μL Perm/Wash buffer including a brilliant violet buffer (BD Pharmingen Stain Buffer, BD Biosciences) for 30 min at 4 °C in the dark. 

Single-colour compensation was performed using 25 μL of compensation beads (UltraComp eBeads and ArC™ Amine Reactive Compensation Bead Kit for LIVE/DEAD compensation, both from Thermo Fisher Scientific), following the manufacturer´s instructions. Acquisition of the samples was performed using an LSR2/LSR Fortessa flow cytometer and FACSDiva Software V.6.1.3 (Becton Dickinson, Frankfurt, Germany). Analysis was performed with the software FlowJo 10.7.0 (FlowJo LLC, Ashland, OR, USA). The gating strategy for flow cytometric analysis is shown in the Appendix A.

### 5.5. Serum SARS-CoV-2 Neutralization Activity

The measurement of neutralizing antibodies (NAb) was performed with a competitive immunoassay (SARS-CoV-2 NAb, YHLO Biotechnology) using labelled recombinant angiotensin-converting enzyme-2 to competitively bind to the microparticle-coupled recombinant SARS-CoV-2 receptor-binding domain, used according to the manufacturer’s instruction on the automated iFlash 1800 platform.

## Figures and Tables

**Figure 1 microorganisms-11-01562-f001:**
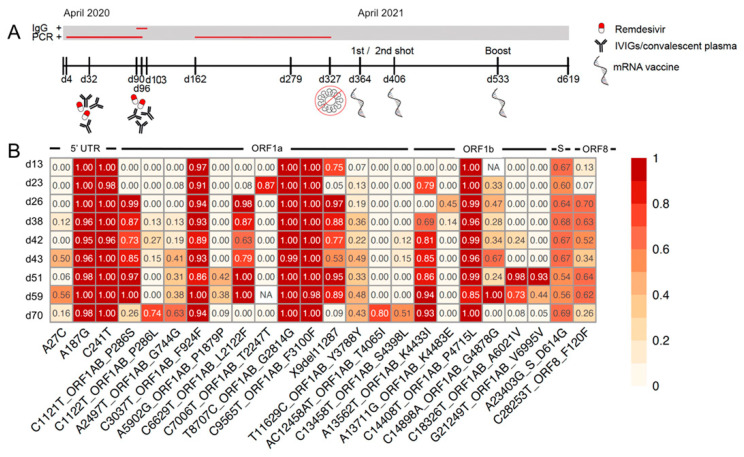
Clinical course and SARS-CoV-2 genome sequencing. (**A**) Overview of the clinical and virological course of COVID-19 in an immunocompromised individual. (**B**) Heatmap showing the longitudinal development of all SARS-CoV-2 variants identified via whole-genome sequencing at a frequency of >40% in at least one of the patient’s nasopharyngeal swab samples. Earliest samples are displayed at the top, with the latest at the bottom. Colour intensities represent variant frequencies from 0 to 1. IVIGs = intravenous immunoglobulins, NA = not available (i.e., no confident variant calls at this position), d = days after infection, UTR = untranslated region, ORF = open reading frame, S = spike gene.

**Figure 2 microorganisms-11-01562-f002:**
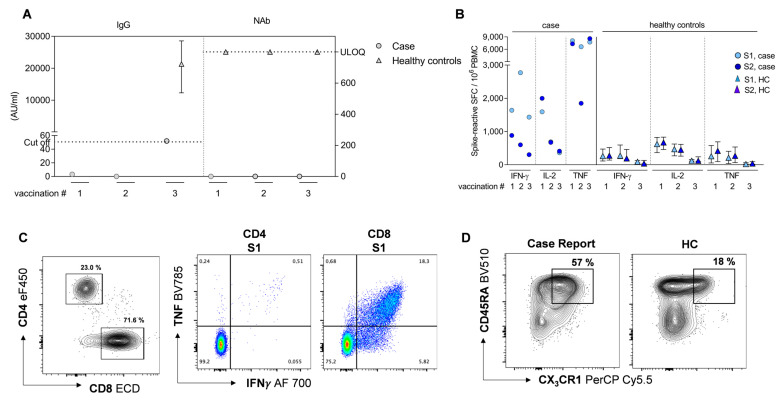
Characterization of SARS-CoV-2-specific humoral and cellular immunity after SARS-CoV-2 infection and BNT162b2 mRNA vaccinations. (**A**) Anti-SARS-CoV-2 IgG levels and virus-neutralization activity (NAb) (case values represented as circles, HC median values represented as triangles with interquartile range). (**B**) Frequencies of SARS-CoV-2 spike-reactive IFN-γ-, IL-2-, and TNF-secreting cells (case, circle) upon re-stimulation of PBMCs with spike S1 (light blue) and S2 (dark blue) peptide pools, as compared to a vaccinated control group (HC, triangle; vaccination #1, n = 50; vaccination #2, n = 20; vaccination #3, n = 9) of non-immunocompromised convalescents with SARS-CoV-2 infection (median with interquartile range shown for HC). (**A**,**B**) Vaccination #1 = 14 days post-1st-COVID-19-vaccination, vaccination #2 = 14 days post-2nd-vaccination, vaccination #3 = 3 months (case) or 14 days (HC) post-3rd-vaccination. (**C**) Frequencies of CD4+ and CD8+ T-cells (left panel) and spike S1-reactive IFN-γ- and TNF-producing CD4+ and CD8+ T-cells (right panel), directly determined ex vivo three months after 3rd vaccination via flow cytometry using intracellular cytokine staining. (**D**) Phenotypic characterization via flow cytometry demonstrates increased expression of CX3CR1 on CD45RA+ CD8 T-cells, relative to a representative HC subject. Representative plots depict total CD8 T-cells. ULOQ = upper limit of quantification. SFC = spot-forming cells. HC data partially published in [8].

## Data Availability

The SARS-CoV-2 viral genome sequences used in this study were deposited in and are freely available via GISAID (https://gisaid.org) (accessed on 25 May 2023) under the following identifiers: EPI_ISL_887264, EPI_ISL_887206, EPI_ISL_887200, EPI_ISL_887205, EPI_ISL_887209, EPI_ISL_887168, EPI_ISL_887191, EPI_ISL_887314, EPI_ISL_887255.

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
