# Peer review of "T-Cell-Dominated Immune Response Resolves Protracted SARS-CoV-2 Infection in the Absence of Neutralizing Antibodies in an Immunocompromised Individual"

_microorganisms, 2023, doi:10.3390/microorganisms11061562_

Round 1

Reviewer 1 Report

The manuscript by Till Bunse et.al. described a study of an immunocompromised patient with a protracted SARS-CoV-2 infection that was eventually cleared in the absence of a humoral neutralizing SARS-CoV-2 antibody response. They followed this patient for about two years, performed SARS-CoV-2 whole genome sequencing on the nasopharyngeal swabs and SARS-CoV-2 specific T-cell profiling on the PBMC. The results of SARS-CoV-2 whole genome sequencing revealed that certain mutations increased their frequencies along the infection course, but none of them were located in regions subject to immune pressure (i.e. the spike gene) because of the patient’s lack of antibody-mediated immune response. Due to the absence of B-cell responses from the patient, SARS-CoV-2 specific T-cell profiling showed that dominant spike S1-specific CD8 T-cell response, which was markedly higher than that in health control group, played major roles in controlling SARS-CoV-2 infection eventually.

However, I have following questions about this study:

1.     First, throughout the course of monitoring this patient, what are the viral loads at different time points? At least, the Ct values of the SARS-CoV-2 specific real-time PCR should be reported. What is the reason of “Viral sequencing beyond day 70 did not yield high quality sequences and was, therefore, not included in the analysis.”(line 116-117)? 

2.     Figure 1 (on page 3) shows that RDV and IGIVs were given on d32. However, even in the days immediately after d32, IgG remained negative. What is the reason for that? We can see IgG+ after d90 treatment. 

3.     Line 106-108: “Over the course of 58 days of continued SARS-CoV-2 PCR-positivity, several variants appeared to increase in frequency.”. Can authors discuss the possible reasons for this? It would be valuable to understand the dynamics of the viral population during the prolonged infection of immunocompromised patient.

4.     Figure 1 (on page 3) and line 114-115: “variants of highly fluctuating frequency were also seen”. Why are there fluctuations at these sites? 

5.     Figure 2C (on page 5): Is there corresponding data from a health control group for comparison? 

6.     Line 199-200: “we cannot exclude a reinfection in our patient, but deemed this unlikely not least due to the low local incidence”. However, considering that the SARS-CoV-2 PCR was positive at that time, it should be possible to amplify specific regions of the SARS-CoV-2 genome that can distinguish between different strains.

7.     There is an extra “T” at the beginning of the Introduction (line 40), which should be removed. 

Reviewer 2 Report

The submitted manuscript is a high-quality case report describing how a chronic infection SARS-CoV-2 was cleared in immunocompromised patient. The authors of the manuscript are making convincing case that vaccination stimulated a T-cell response in the patient that eventually promoted the virus clearance. This is an important topic because there are still a lot of gaps in understanding the functioning of T-cell immunity in the fight against viral infections in general and SARS-CoV-2 in particular. Filling in these gaps will undoubtedly help to better understand how to deal with COVID-19 and how to make vaccines more effective. In addition, the information presented in the manuscript may provide insight into the mechanism by which immunocompromised patients can better cope with chronic SARS-CoV-2 viral infection. It provides hope and healthcare direction for immunocompromised and chronically sick patients.

The manuscript is informative, written logically and clearly. However, it is worth noting minor shortcomings, removing which the authors will help the reader to better and faster understand and perceive the material presented.

1. In order to improve the understanding of this work by the readers of the manuscript, I would like to recommend the authors to make a general scheme (Timeline) of events in the treatment of the patient in question. The manuscript will greatly benefit if it includes a timeline of clinical manifestation of disease, tests taken, days when the treatment or vaccination was given and so on. I suggest including real dates, because it can help to understand when the patient was first infected.

2. The description of the clinical case (section 2. Case Report) lacks information about when the virus was cleared and how the patient felt. The abstract of the work briefly reports this, but I did not see this information in the description of the clinical case. Does the patient still have signs of a long COVID-19, and if so, is there a trend towards their improvement.

3. Line 65. “During the first wave of the pandemic, a 69-year-old woman was admitted to the”. As I mentioned before, the date of the admission or of first signs of clinical manifestation of disease would be helpful for the reader.

4. Line 106. “…infected with the dominant viral strain at the time (Pangolin lineage B.1).” Authors must be more specific what lineage of the virus they sequenced. There is no such lineage as Pangolin B.1. If the sequence was partial and can’t reveal the lineage than date of hospital admission or date of the disease manifestation must be provided, so a reader can check what dominant virus variants were circulating at the time.

5.Lines 92-93 “In the following months, the patient received three vaccinations with the mRNA vac- 92 cine BNT162b2 (Comirnaty®, BioNTech-Pfizer) in line with the recommendations of the German national authorities.” I am repeating myself, but this information must be more specific and presented in the timeline.

6.  I highly recommend for the manuscript authors to upload the sequences of the SARS-CoV-2 variants that they obtained to NCBI (NIH) https://submit.ncbi.nlm.nih.gov/sarscov2/. The sequence variants along with the case report are highly valuable for researchers of chronic SARS-CoV-2 infections in immunocompromised patients. These researchers may first find the sequences and then find the publication. The sequence(s) in the database is another entry point to this valuable research.
